# Nutritional Outcome in Home Gastrostomy-Fed Children with Chronic Diseases

**DOI:** 10.3390/nu11050956

**Published:** 2019-04-26

**Authors:** Cecilia Martínez-Costa, Caterina Calderón, Lilianne Gómez-López, Soraya Borraz, Elena Crehuá-Gaudiza, Consuelo Pedrón-Giner

**Affiliations:** 1Department of Pediatrics, School of Medicine, University of Valencia, Hospital Clínico Universitario of Valencia, Avenida Blasco Ibáñez 15–17, 46010 Valencia, Spain; lilian.ne.gomez@gmail.com; 2Gastroenterology and Nutrition Unit, Hospital Clínico Universitario, Avenida Blasco Ibáñez 17, 46010 Valencia, Spain; elenacrehua@gmail.com; 3Department of Clinical Psychology and Psychobiology, Faculty of Psychology, University of Barcelona, 08035 Barcelona, Spain; ccalderon@ub.edu; 4Department of Pediatrics, Hospital de Dénia, Partida de Beniadtlá s/n. Denia, 03700 Valencia, Spain; soraya.borraz@marinasalud.es; 5Gastroenterology and Nutrition Unit, Hospital Infantil Universitario Niño Jesús, Menéndez Pelayo 65, 28009 Madrid, Spain; consuelocarmen.pedron@salud.madrid.org

**Keywords:** gastrostomy, home enteral nutrition, home nutritional support, chronic illness, nutritional support in children

## Abstract

The aim of the study was to assess the anthropometric outcomes after gastrostomy tube (GT) placement in children with chronic diseases and the influence of primary diagnosis, age, and nutritional support. A longitudinal, multicenter, and prospective study was performed evaluating 65 children with GT feeding and chronic diseases (61.5% with neurological disease). Each child was evaluated three times (at baseline and at 6 and 12 months after GT placement) and the following data was collected: primary diagnosis, age at GT placement, anthropometry, and feeding regime. Repeated measures ANOVA were used to analyze the main effects (intra and intergroup) and the interactions effects on weight gain and linear growth at 6 and 12 months after GT placement. All patients significantly improved their body mass index (BMI)-for-age z-score (*p* < 0.001) and height-for-age z-score (*p* < 0.05) after 6 and 12-month of follow-up. BMI gain increased significantly the first 6 months, followed by a plateau, while height followed a linear trend. Children with GT placement before 18 months old experienced an accelerated growth rate during the first 6 months post-GT. This technique showed the effectiveness of GT placement improving nutritional status and growth catch up regardless of their primary diagnosis and the type of nutritional support.

## 1. Introduction

Pediatric patients with chronic diseases often suffer growth delay and restriction on their nutritional status linked to feeding issues, high nutritional requirements, and/or an inadequate caloric intake, which means they need specific nutritional support to prevent undernutrition and growth failure [1,2,3,4,5]. Home enteral nutrition (HEN), specifically by gastrostomy tube (GT), has been proved to be a secure technique for these patients that improves or reverts an undernourishment state and simplifies their care, reduces hospitalization length, and decreases respiratory and infectious complications (i.e., aspirative pneumonia due to severe disphagia and subsequent infection) [1,6,7,8]. Neurological and muscular disorders, congenital heart disease, respiratory diseases such as cystic fibrosis, inherited metabolic, and oncologic diseases are chronic conditions that frequently demand this nutritional support [7,8,9,10,11]. In these patients, GT feeding, especially in highly dependent children, contribute to better outcome of the main disease and prolong their survival rate, with this being particularly the case for children suffering from severe neurological disability [11,12,13]. Moreover, this nutritional support may also favor the patients’ global improvement, increasing their own and their carers’ quality of life [10,14,15]. In preliminary studies, the main carers reported high satisfaction with GT feeding, and would have accepted its earlier implementation if they had foreseen its benefits [16,17]. They also perceived an improvement in their child’s nutritional status higher than that quantified by the anthropometric assessment [18].

Despite most healthcare professionals noticing this nutritional improvement, literature has rarely conveyed the effects (prospectively and longitudinally) of GT feeding in the growth rate and development of children with chronic diseases [6,15], nor the influence of different factors related to patients and its nutritional support mode. For this reason, our objectives were: First, to analyze the anthropometric outcomes after GT placement in children with chronic diseases. Second, to examine whether there were differences in the nutritional evolution according to the following variables: primary diagnosis, age and nutritional status at GT placement, feeding regimen, and infusion system. 

## 2. Patients and Methods

### 2.1. Study Design

An observational, prospective, and longitudinal study was carried out at the Paediatric Gastroenterology and Nutrition Units of two public hospitals in Spain. The sample comprised pediatric patients with chronic diseases for whom HEN by GT feeding was prescribed during the follow-up (due to the impossibility of oral feeding and/or the inability to cover energetic needs). Exclusion criterion was the refusal of the children’s carers to participate in the survey. The study protocol was approved by the Ethics Committee of each hospital in accordance with the Declaration of Helsinki of 1964, revised in Seoul in 2008. Written parental informed consent was obtained. 

### 2.2. Participants

During the study phase, 98 patients with GT placement for HEN were recruited. Thirty-three patients were later excluded due to the impossibility of completing a longitudinally study in the three phases (6 patients with a skeletal deformity because their height could not be assessed precisely, 7 patients suffering a genetic disorder that affected their growth, and 20 patients because at least one anthropometric measure was missed). The final sample comprised 65 children (36 boys and 29 girls), aged between 8 months and 16.2 years old (mean 3.1; SD 2.4). The main diagnostic categories were: 40 patients (61.5%) with neurological disease, 14 (21.5%) with cardio-respiratory disease, 6 patients (9.3%) were affected by an inherited metabolic disease, and a miscellaneous group of 5 patients (7.7%): 2 children with swallowing disorders, 2 with digestive disorders, and 1 with an oncologic disease. The majority of patients (61.5%) with a neurological disease suffered cerebral palsy (CP), with severe motor impairment (Gross Motor Function Classification System -GMFCS- grade IV and V) [19]. To assess the influence of early GT implementation, children were divided into two groups: under 18 months, and over 18.1 months. 

### 2.3. GT Placement and Nutrition Support Mode

GTs were placed through percutaneous endoscopic gastronomy (PEG) in 52 patients (50 using the pull-through technique and 2 via gastropexy) and with open surgery in 13 patients [20,21]. In those patients where the PEG-tube was placed using a standard pull-through technique, the tube was replaced with low profile button devices (Mic-Key^®^) 5 to 6 months later.

To assess the nutritional support, the following variables were analyzed: type of food received through GT (homemade meals vs. enteral formulas vs. combined food), feeding regime, and infusion system. Patients were classified into the “combined food” group when they were considered to consume both types of food in similar proportions. For feeding regime, the distinction was made between intermittent bolus feeding and cyclic infusion. In order to describe the infusion system, it was subdivided into three categories: syringes, pump, or both. 

### 2.4. Anthropometric Evaluation

Each child went through anthropometric assessment three times: once at the baseline, and two more at 6 and 12 months after GT placement. Weight and height were collected and body mass index (BMI) was calculated as weight (kg) divided by height squared (m^2^). Each child’s weight and height were measured according to a standard technique and with suitable equipment for the age: clinical scale or Seca^®^ baby scale (models 755/334) and infant measuring table or Holtain^®^ stadiometer. In old patients with severe motor impairment, the supine length was assessed using Holtain Harpenden Anthropomer (Holtain Ltd.^®^, Felin-y-Gigfran, Crosswell, UK). Measurements of weight and height were transformed to age- and sex-specific BMI-for-age (Z_BMI/A_) and height-for-age (Z_H/A_) z-scores using WHO growth standards [World Health Organization -WHO- Anthro Software (version 3.2.2, January 2011, WHO, Geneva, Switzerland) for children under 5 and WHO Anthro Plus Software (version 3.2.2, January 2011, WHO, Geneva, Switzerland) for children 5–19 years [22,23]. In order to categorize nutritional status, cut-off points established by WHO were applied, considering both Z_BMI/A_ and Z_H/A_. Applying Z_BMI/A_ children were classified as follows: a score ≥ −2 and ≤ +1 SD was considered normal nutritional status (being > +1 SD overweight and > +2 SD obesity); and a score < −2 was considered acute undernutrition (low-weight-for-height or wasting), delimiting the following degrees: moderate when score was <−2 and ≥ −3 and severe undernutrition if score was <−3. Z_H/A_ between ≥ −2 and ≤ +2 SD was classified as normal. Stunting or chronic undernutrition (low-height-for-age) was considered in the case of a Z_H/A_ z-score that was <−2 and normal Z_BMI/A_ [24]. Children with both Z_BMI/A_ and Z_H/A_ <−2 were categorized as wasting and stunting simultaneously. 

### 2.5. Statistical Analysis

Descriptive statistics were used to analyze the study variables for each of the three assessments. Multivariate analysis was conducted to assess the anthropometric outcomes after GT placement in children with chronic diseases. A repeated measures ANOVA was used to analyze the main effects (intra and inter group) and the interaction effects on weight gain (Z_BMI/A_) and linear growth (Z_H/A_) at 6 and 12 months after GT placement. To examine within-subjects change in anthropometric outcomes across time, separate repeated-measures ANOVA were conducted on the measures of Z_BMI/A_ and Z_H/A_ at 6 and 12 months after GT placement, with primary diagnostic, age of child, nutritional status, feeding regime, and infusion system. Before the ANOVA analysis, Mauchly’s test of sphericity and Levene’s test of equality of error variances were applied. Post hoc pairwise comparisons were done using the Bonferroni procedure and/or contrast Tamhane’s T2 multiple comparison test as appropriate. The effect size was analyzed with partial eta-squared to compare patients’ pre-treatment measures with post-treatment measures. For statistical analysis, the Statistical Package for the Social Sciences (SPSS, INC., Chicago, III) was used, version 23.0. For all the tests carried out, bilateral statistical significance was set at *p* ≤ 0.05. 

## 3. Results

### 3.1. Anthropometric Outcomes following GT Placement

The nutritional categorization before GT implantation revealed the following nutritional status: 20 children (30.8%) were normal and 45 (69.2%) exhibited undernutrition; in particular, 6 (9.3%) showed acute undernutrition, 22 (33.7%) chronic undernutrition, and 17 (26.2%) both stunting and wasting. Graded acute undernutrition showed that 8 patients (12.3%) suffered moderate undernutrition and 14 patients (21.4%) severe undernutrition.

The results of Z_BMI/A_ and Z_H/A_ z-score at baseline, 6 months, and 12 months are represented in Table 1. Globally, the longitudinal anthropometric evaluation following GT placement revealed significant improvements in Z_BMI/A_ and Z_H/A_. All of them experienced a statistically significant increase of their Z_BMI/A_ (F = 22.344, *p* = 0.0001), particularly on the first 6 months, showing a *quadratic trend* (MD = −0.81, *p* = 0.001, IC95% = −0.48 to 1.14) followed by a phase of stabilization between months 6 to 12 (MD = 0.09, *p* = 1.000, IC95% = −0.25 to 0.43). Height improved in all patients (F = 6.078, *p* = 0.016) linearly increasing for the first 6 months (MD = −0.22, *p* = 0.031, IC95% = −0.43 to 0.02) and between months 0 to 12 (MD = −0.27, *p* = 0.049, IC95% = −0.54 to 0.00). 

Descriptive statistics of Z_BMI/A_ and Z_H/A_ for the three times grouped by the study variables are showed in Table 2 and Figure 1 and Figure 2. 

### 3.2. Anthropometric Outcomes According to Primary Diagnosis, Age, and Nutrition Status at GT Placement

Repeated-measures ANOVA were used to analyze anthropometric outcomes according to the following variables: primary diagnosis, age, and nutritional status at GT placement, and nutritional support mode (Table 3, and Figure 1 and Figure 2). 

Anthropometric evolution according to primary diagnosis: Anthropometric variables indicated the within-subjects main effect in Z_BMI/A_ (F = 13.336, *p* < 0.0001) and Z_H/A_ (F = 4.668, *p* = 0.035) across time, therefore nutritional treatment was proved efficacious in all children regardless of primary diagnosis. Post-hoc tests (Bonferroni) revealed that children with a metabolic disease presented a higher Z_BMI/A_ than those with a neurological disease (*p* = 0.001) (Figure 1). 

Anthropometric outcomes according to GT placement age: Likewise, anthropometric measures indicated a significant within-subjects main effect in Z_BMI/A_ (F = 18.858, *p* < 0.0001) and Z_H/A_ (F = 6.492, *p* = 0.013) after one year of HEN. Pairwise comparisons indicated a significant increase in anthropometric measures (Z_BMI/A_ and Z_H/A_) at baseline and 6 months (*p* < 0.01 in both cases), but these were not significant at 6 to 12 months (Figure 1 and Figure 2). These results indicate that the improvement in the BMI was not influenced by the GT placement before or after 18 months. Nevertheless, there was a more pronounced improvement in height (at 6 months) in the children in whom the implantation of the GT feeding was performed before 18 months of treatment. 

Anthropometric evolution according to nutritional status prior to GT placement: Anthropometric measures indicated a significant within-subjects main effect in Z_BMI/A_ (F = 14.928, *p* = 0.001) but not in Z_H/A_ (F = 2.239, *p* = 0.132) after one year of HEN (Table 3). This proves that children with acute undernutrition and those with both stunting and wasting experienced an improvement in BMI that was notoriously higher than that of normal and chronically malnourished children at baseline. 

Anthropometric evolution according to nutritional support mode: Anthropometric evolution was analyzed according to the type of food (homemade meals vs. enteral formula vs. combined), feeding regime (cyclic vs. bolus feeding), and infusion system (pump, syringe, or both). A significant improvement was found in all the patients, with no differences related to the type of feeding, feeding regime, or infusion system. In relation to the type of food, results of the repeated-measures ANOVA performed indicated a significant within-subjects main effect in Z_BMI/A_ (F = 21.194, *p* < 0.0001) and Z_H/A_ (F = 5.426, *p* = 0.023) after one year of treatment, without differences among them. In relation to feeding regime, results indicated a significant within-subjects main effect in Z_BMI/A_ (F = 22.882, *p* < 0.001) and Z_H/A_ (F = 6.480, *p* = 0.013) after one year of treatment. Considering the infusion system, notorious improvements were found in Z_BMI/A_ (F = 22.392, *p* < 0.0001) and Z_H/A_ (F = 8.819, *p* = 0.004). 

## 4. Discussion

These study results evidence that HEN using a GT is an efficacious treatment that improves the nutritional status of children suffering from chronic diseases. All participating patients gradually enhanced their Z_BMI/A_ during the 12 months of treatment, especially for the first 6 months (over 20% of their initial values in most cases). Similarly, other authors have documented positive responses in the first 6 months of treatment [6,25]. Concerning height measures, our results show that almost all patients evidenced a gradual growth improvement. Similarly, in a longitudinal study for 44 children with CP, Sullivan et al. observed improvements in 2005 for all anthropometric parameters between baseline and 6 and 12 months after GT insertion [1]. In addition, Lalanne et al. showed in 2014 in a longitudinal study that this gain was maintained for a long term [26]. Regarding the height outcome, there are certain discrepancies, with studies that similarly agreed that height improves parallel to HEN [27,28], while others concluded growth was stationary and followed no significantly positive trend [7,29]. This different outcome is probably related with non-nutritional factors that can stop growth and with the time of treatment initiation. Considering that in the first 18 months of age is when the growth rate is highest, and therefore the risk of undernutrition and growth arrest is also greater, we hypothesized that in patients with an early GT placement (before 18 months of age), the growth rate could be better improved, compared to those where placement was done later in life. A preliminary study comprising 26 children with GT feeding evidenced better height measures in those children with a precocious GT placement [16]. In our actual sample, we observed a notorious improvement in height but only in the first 6 months post GT placement. Other studies also concluded that reach minimum growth standards are reached more frequently in children with an early GT placement before severe undernutrition has been established [1]. 

Regarding the primary diagnosis, the longitudinal analysis evidenced a positive response for the whole sample, regardless of the main diagnosis. Nevertheless, children with neurological diseases exhibited at baseline had a poor nutritional status compared with others diagnoses (Figure 1 and Figure 2). It must be taken into consideration that most patients with CP had severe motor impairment and the age of GT placement was higher. Other series grouping patients with different diagnosis equally observed an overall improvement [16,27,28], although most studies sampled children suffering from a particular disease, such as CP [1,30], neuromuscular diseases [29], cystic fibrosis [7], chronic kidney diseases [31], and oncological diseases [32]. Concerning the study design, differently from the study conducted by Dispaquale et al. in 2018 in impaired children, we excluded patients with genetic disorders as their growth is determined mainly by factors other than nutritional [25].

Regarding the nutritional status prior to GT placement, we observed that this nutritional support was efficient in reverting undernutrition. However, overall improvement was clinically notorious in patients suffering from acute undernutrition or from a stunted and wasted situation as they gained a considerable amount of weight, a fact similarly observed by other authors [1,16,27,30]. All these findings suggest that the nutritional support must be implemented as soon as possible, particularly in patients suffering from severe chronic diseases, who will need this nutritional support for years or even lifelong. 

Focusing on the anthropometric measures reached in the whole sample by the end of the study, we observed that all children improved their nutritional status but not all of them started from the same baseline conditions. Children that at the beginning of the study suffered from neurological diseases presented more unfavorable conditions compared to the remainder, especially with children with metabolic diseases. These differences persisted throughout the study in both groups. These results suggest that in patients suffering from neurological diseases, HEN comes later in time, commonly reaching a severe state of undernutrition with stopped growth, sometimes irreversibly. The higher the deterioration of the nutritional status, the more difficult it is to recuperate. This finding is consistent with other studies that highlight the vulnerability of these patients [33]. Moreover, it must be noted that in patients with neurological diseases, their families frequently find it difficult to accept GT placement, delaying the procedure, thus it is crucial to properly advise and give adequate support to these families during the decision-making process for GT insertion [16,34,35]. Establishing a proper nutritional follow-up process may prevent this delay and the morbidity and mortality associated with undernutrition [11,30].

In relation to other factors that may influence the anthropometric follow-up, this study found no significant association with the type of food received by GT, nor with the feeding regime or the infusion system. With respect to the effect of the infusion system within subjects, one possible explanation is that the nutritional status probably influences the mode of nutritional support since the most undernourished patients usually have lower tolerance. Recently, a study conducted in impaired children after GT placement showed that enteral nutrition with a standard polymeric formula proved to be efficient in reversing undernutrition but they did not compare with other modes of feeding [25]. Latest studies have shown that blenderized food has been steadily gaining in popularity among parents and caregivers of tube-fed children [36,37]. In a survey to assess the prevalence of this practice, about 9% of pediatric patients used blenderized food for an average of 71% of their total daily nutrition intake [38]. To our knowledge, no other study has assessed feeding regime or infusion system, although further investigations have to consider its association with complications and quality of life standards. According to our own experience, the nutritional support should be selected individually for each patient with the aim of not only fulfilling the patient’s hydric and nutritional needs, but also securing the best possible adaptation to the children and their families’ life. 

The greatest limitation in this report was the lack of complete data for some patients, thereby causing the initial sample to be reduced. This lack of data was a consequence of difficulties in these patients’ follow-up and, mainly, the complexity of obtaining precise measurements for the children’s basic parameters, such as height. In addition, the follow-up included in this study was limited to one year, therefore it is not possible to conclude whether BMI and height increase or stabilize over a longer period. Other studies that included anthropometric measurements that assessed the corporal composition of arm circumference and skinfolds (triceps and subscapular) and biological parameters would be of significant medical interest. 

## 5. Conclusions

In conclusion, taking in consideration the limitations of the study, our results evidence that GT feeding improved the nutritional status of children with chronic diseases, regardless of their primary diagnosis and the type of nutritional support. Moreover, in those children with early GT implantation, that is, before 18 months of age, the growth rate was higher during the first 6 months post-GT, especially in children suffering from acute undernutrition in which the anthropometric improvement was notorious.

## Figures and Tables

**Figure 1 nutrients-11-00956-f001:**
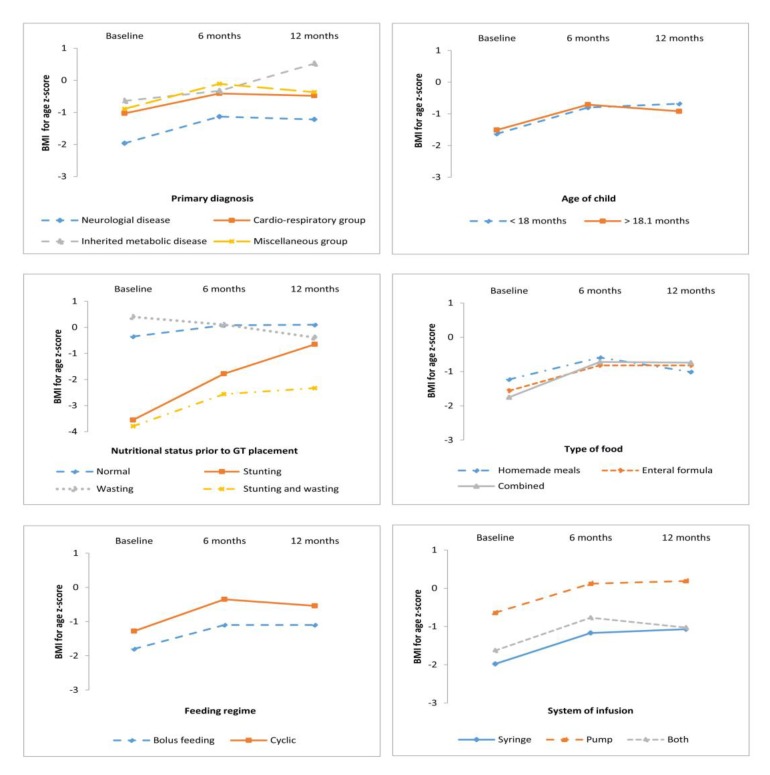
BMI-for-age z-score according to analyzed variables.

**Figure 2 nutrients-11-00956-f002:**
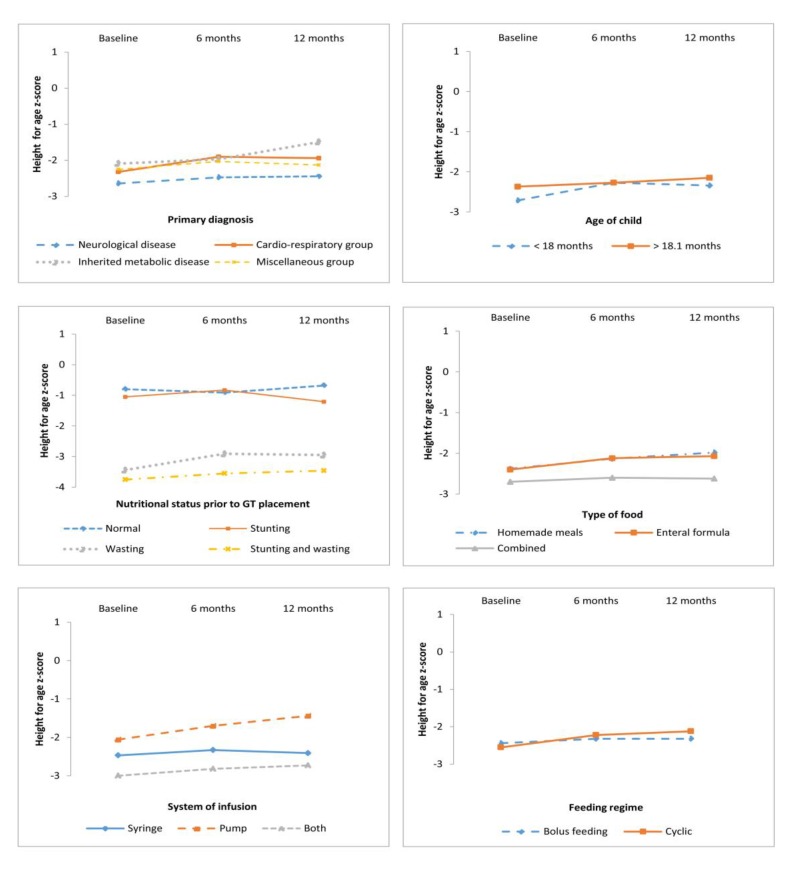
Height-for-age z-score according to analyzed variables.

**Table 1 nutrients-11-00956-t001:** Anthropometric outcomes following GT placement.

**Anthropometry** **z-score ***	***n***	**Baseline** **M ± SD**	**6 months** **M ± SD**	**12 months** **M ± SD**	**F**	***p***
**BMI-for-age**	65	−1.56 ± 1.95	−0.74 ± 1.62	−0.83 ± 1.42	22.344	0.0001
**Height-for-age**	65	−2.50 ± 1.62	−2.27 ± 1.50	−2.22 ± 1.64	6.078	0.016
**Anthropometry** **z-score ***	**Comparison ***		**Mean difference**	***p***	**95% CI of the difference**	
					**Lower**	**Upper**
**BMI-for-age**	Baseline vs. 6 months		−0.81	0.001	−0.48	1.14
	Baseline vs. 12 months		−0.72	0.001	−1.20	0.24
	6 months vs. 12 months		0.09	1.000	−0.25	0.43
**Height-for-age**	Baseline vs. 6 months		−0.22	0.031	−0.43	−0.02
	Baseline vs. 12 months		−0.27	0.049	−0.54	0.00
	6 months vs. 12 months		−0.05	1.000	−0.24	0.15

* Binary comparisons with the Bonferroni adjustment to analyze the differences between the HEN pre-treatment and the follow-up at 6 months and 12 months in the parameters BMI-for-age z-score and height-for-age z-score.

**Table 2 nutrients-11-00956-t002:** Means and standard deviation of BMI-for-age z-score and height-for-age z-score for the three times grouped by the study variables.

Variables	*n*	%	BMI-for-Age z-Score			Height-for-Age z-Score		
			Baseline	6 Months	12 Months	Baseline	6 Months	12 Months
			M (SD)	M (SD)	M (SD)	M (SD)	M (SD)	M (SD)
**Primary Diagnosis**								
**Neurological disease**	40	61.5	−1.96 (2.12)	−1.13 (1.68)	−1.22 (1.51)	−2.64 (1.76)	−2.47 (1.56)	−2.44 (1.45)
Cardio-respiratory disease	14	21.5	−1.03 (1.43)	−0.41 (1.55)	−0.48 (1.01)	−2.32 (1.39)	−1.90 (1.47)	−1.94 (2.10)
Inherited metabolic disease	6	9.3	−0.64 (1.04)	−0.33 (0.78)	0.52 (0.50)	−2.09 (1.81)	−1.97 (1.65)	−1.49 (2.11)
Miscellaneous group ^a^	5	7.7	−0.89 (2.16)	0.11 (1.19)	−0.37 (1.08)	−2.25 (1.01)	−2.03 (0.83)	−2.13 (1.01)
**Age of Child**								
<18 months	24	36.9	−1.63 (2.00)	−0.80 (1.58)	−0.68 (1.26)	−2.71 (1.50)	−2.27 (1.49)	−2.34 (1.61)
>18.1 months	41	63.1	−1.51 (1.94)	−0.71 (1.67)	−0.92 (1.51)	−2.37 (1.70)	−2.27 (1.53)	−2.15 (1.67)
**Nutritional Status**								
Normal nutritional status	20	30.8	−0.35 (0.99)	0.08 (1.01)	−0.10 (1.01)	−0.80 (0.93)	−0.91 (0.82)	−0.68 (1.16)
Acute malnutrition (wasting)	6	9.3	−3.55 (1.27)	−1.78 (1.07)	−0.65 (0.81)	−1.05 (0.66)	−0.83 (0.96)	−1.21 (0.75)
Chronic malnutrition (stunting)	22	33.7	0.40 (1.06)	0.10 (0.95)	−0.39 (1.01)	−3.44 (1.00)	−2.91 (1.01)	−2.95 (0.96)
Stunting and wasting	17	26.2	−3.76 (1.30)	−2.56 (1.26)	−2.33 (1.40)	−3.75 (0.92)	−3.55 (1.10)	−3.46 (1.44)
**Type of Food**								
Homemade meals	13	20.0	−1.23 (1.82)	−0.59 (1.58)	−1.01 (1.48)	−2.38 (2.34)	−2.13 (1.95)	−1.98 (2.20)
Enteral formula	32	49.2	−1.56 (1.91)	−0.82 (1.64)	−0.82 (1.37)	−2.40 (1.50)	−2.12 (1.36)	−2.07 (1.43)
Combined	20	30.8	−1.75 (2.15)	−0.72 (1.69)	−0.74 (1.51)	−2.70 (1.29)	−2.60 (1.42)	−2.62 (1.56)
**Feeding Regime**								
Bolus feeding	34	52.3	−1.80 (2.20)	−1.10 (1.73)	−1.10 (1.53)	−2.44 (1.92)	−2.32 (1.66)	−2.32 (1.70)
Cyclic	31	47.7	−1.28 (1.63)	−0.35 (1.43)	−0.54 (1.24)	−2.55 (1.25)	−2.22 (1.34)	−2.12 (1.59)
**System of Infusion**								
Syringe	34	52.3	−1.98 (2.24)	−1.17 (1.75)	−1.07 (1.45)	−2.47 (1.94)	−2.33 (1.77)	−2.41 (1.93)
Pump	17	26.2	−0.64 (1.28)	0.12 (1.09)	−0.19 (1.12)	−2.06 (1.23)	−1.70 (0.97)	−1.44 (1.13)
Combined	14	21.5	−1.63 (1.53)	−0.77 (1.51)	−1.03 (1.51)	−3.06 (1.01)	−2.82 (1.10)	−2.73 (1.05)

^a^ Swallowing disorder (*n* = 2), digestive disorder (*n* = 2), oncologic disease (*n* = 1).

**Table 3 nutrients-11-00956-t003:** Summary of the effects for the anthropometrics variables (ANOVA of repeated measures) over time.

Variable	Source	BMI-for-Age z-score		Height-for-Age z-score	
		F	*p*	F	*p*
**Primary diagnosis**	Within-subjects effects	13.336	**<0.0001**	4.668	**0.035**
	Interaction	0.233	0.873	0.440	0.725
	Between-groups effects	2.900	**0.042**	0.612	0.610
**Age of GT placement**	Within-subjects effects	18.858	**<0.0001**	6.492	**0.013**
	Interaction	0.816	0.370	4.600	**0.036**
	Between-groups effects	0.001	0.974	0,195	0.660
**Nutritional status prior GT placement**	Within-subjects effects	14.928	**0.001**	2.329	0.132
	Interaction	9.946	**0.001**	1.141	0.340
	Between groups effects	38.603	**0.001**	36.611	**0.001**
**Type of food**	Within-subjects effects	21.194	**<0.0001**	5.426	**0.023**
	Interaction	0.975	0.383	0.691	0.505
	Between-groups effects	0.035	0.965	0.189	0.665
**Feeding** **regime**	Within-subjects effects	22.882	**0.001**	6.480	**0.013**
	Interaction	0.007	0.932	1.954	**0.664**
	Between-groups effects	2.677	0.107	0.189	0.664
**System of infusion**	Within-subjects effects	22.392	**<0.0001**	8.819	**0.004**
	Interaction	0.526	0.593	2.435	0.096
	Between-groups effects	0.029	0.865	2.254	0.114

Bold values indicate the significant at 5% level. GT: Gastrostomy tube.

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
