# Peer review of "Nutritional Outcome in Home Gastrostomy-Fed Children with Chronic Diseases"

_nutrients, 2019, doi:10.3390/nu11050956_

Reviewer 1 Report

Line 61. Study design. Clear criteria of inclusion and exclusion are lacking.

Line 68: participants. Too many drop outs (about 30% of the patients included). That shows a not reliable identification of inclusion and exclusion criteria.

Line 74 an following: main diagnostic categories were: 40 patients (61.5%) with neurological disease, 14 (21.5%) with 74 cardio-respiratory disease, 6 patients (9.3%) were affected of inherited metabolic disease, and a 75 miscellaneous group of 5 patients (7.7%): 2 children with swallowing disorders, 2 with digestive 76 disorders, and 1 with oncologic disease. The majority of patients (61.5%) with neurological disease 77 suffered cerebral palsy (CP), with severe motor impairment (Gross Motor Function Classification 78 System -GMFCS- grade IV and V) [19]. To assess the influence of early GT implementation, children 79 were divided into two groups: under 18 months, and over 18.1 months. The authors non clearly defined the underlying disease. They also report that 61.5% were neurological patients and that the majority of neurological patients were affected by cerebral palsy (61.5%). Therefore all patients with neurological disease were cerebral palsy? But they reported that 77 patients suffered from cerebral palsy with percentage of 78.5 of the group and a number higher that the total patients included in the study   The two patients with swallowing disorders which disease had?  The Authors should also clarify which type of enteral formulation they used

Line 93: Anthropometric evaluation 93

Each child went through anthropometric assessment three times: once at the baseline, and two 94 more at 6 and 12 months after GT placement. Weight and height were collected and body mass index 95 (BMI) was calculated as weight (kg) divided by height squared (m2). How did you measure the height in neuro-disabled patients?. You have to clarify the mode to assess height in this category of patients, for whose is often very difficult to evaluate this parameter.

Statistical analysis. Multivariate analysis is lacking.  There were too many variables that could impact on the final outcome and that should be correctly analyzed  (In particular this evaluation should be applied to the data reported in Table3. In this Table you reported that system of infusion  has impact on nutritional status? It’s possible that system of infusion could blind some other factors)

 Line 122-124

Anthropometric outcomes following GT placement 123

The nutritional categorization before GT implantation revealed the following nutritional status: 124 20 children (30.8%) were normal and 45 (69.2%) exhibited undernutrition; in particular, 6 (9.3%) 125 showed acute undernutrition, 2 Why in patients with normal nutritional status EN has been performed? Also in presence of swallowing disorders generally the nutritional status could be improved

Author Response

We would like to thank you in advance for giving us the opportunity to revise the manuscript. The complete response to the Reviewer 1 is included in the attached file. All changes following your recommendations have been highlighted in yellow in the manuscript.

looking forward for your replay

Reviewer 2 Report

This manuscript “Nutritional outcomes in home gastrostomy-fed children with chronic diseases” aims to identify nutrition outcomes in children following gastrostomy placement.  The authors should be commended on their presentation of such clinically relevant data reporting the important topic of nutrition outcomes and identifying trends among subpopulations of patients receiving g-tubes.  Additional clarifications would be helpful in enhancing the value of the manuscript: 

Major revisions:

Anthropometric evaluation, pg 3, line 99-102.  WHO uses wt/length z-scores under the age of 2 yrs and BMI z-scores over 2 yrs of age.  Please clarify use of BMI in children under 2 years and standard use of wt/length and how this affects results. 

Anthropometric evaluation, pg 3, line 104.  “> -2  and < +1 SD”  According to pediatric malnutrition criteria, this should be > -1 to < +1 SD.  Please explain use of your criteria and its effect on results.

Patients and Methods, pg 2, line 80.  “under 18 months… “  Please explain rationale for age division. 

Minor revisions:

Introduction, pg 1, line 42.  “… decreases respiratory and infectious complications.”  Please clarify what type of complications you are referring to (i.e. respiratory complications due to severe dysphagia and subsequent aspiration).

Discussion, pg 7, line 187 and pg 8 line 218.  “efficient treatment” Please clarify.  Study does not report on the efficiency of the gastrostomy feedings.  Did authors intend to say “efficacious”? 

Discussion, pg 8, line 226-228.  Although the authors discuss the need for future research to look at corporal composition, recommend authors comment along with findings in participants with neurologic impairment, as it is know that BMI is not a good indicator of nutritional status in this population. 

Author Response

We would like to thank you in advance for giving the opportunity to revise our manuscript. 

All changes following your recommendations have been highlighted in yellow in the manuscript.

The complete answer (file author-coverletter) and supplementary Annexe (Report notes) are included in attached files.

Looking forward for your replay

Round  2

Reviewer 2 Report

Accept in present form.